# Analysis of Children’s Physical Characteristics Based on Clustering Analysis

**DOI:** 10.3390/children8060485

**Published:** 2021-06-07

**Authors:** Eunjung Kim, Yumi Won, Jieun Shin

**Affiliations:** 1Division of Sports Science, Myongji University, Yongin-si 17058, Korea; ikoala@mju.ac.kr; 2Department of Exercise Rehabilitation, Daewon University College, Jecheon-si 27135, Korea; 3Department of Biomedical Informatics, College of Medicine, Konyang University, Daejeon-si 35365, Korea

**Keywords:** children, physical fitness, cluster

## Abstract

This study assessed the physical development, physical fitness (muscular endurance, muscular strength, flexibility, agility, power, balance), and basal metabolic rate (BMR) in a total of 4410 children aged six (73–84 months) residing in Korea. Their physical fitness was visually classified according to the physical fitness factor and—considering that children showed great variations in the physical fitness criteria depending on their physique and body composition—the study aimed to assess characteristics such as physique and BMR, the precursor for fat-free mass, based on the physical health clusters selected through a multivariate approach. As a result, the physical health clusters could be subdivided into four clusters: balance (1), muscular strength (2), low agility (3), and low physical fitness (3) cluster. Cluster 1 showed a high ratio of slim and slightly slim children, while cluster 2 had a high proportion of children that were obese, tall, or heavy, and had the highest BMR. We consider such results as important primary data for constituting physical fitness management programs customized to each cluster. It seems that it is necessary to have a multidirectional approach toward physical fitness evaluation and analysis methodologies that involve various physical fitness factors of children.

## 1. Introduction

The number of overweight or obese toddlers and children aged below 5 years was 124 million worldwide as of 2017 [1]. This is a tenfold increase during the past 40 years, and the number has reached approximately 213 million [1]. The Ending Childhood Obesity report stated that one of the public health problems the world should resolve in the 21st century is child obesity [2].

The reason why child obesity is at the center of concern is that the obesity of toddlers and children is related to early death. Obesity is a disease with which people have high nutritional intake but do not consume energy enough, cumulating excessive body fat as the imbalance between energy intake and consumption continues [3]. In other words, lack of physical activity leads to overweight and obesity, which becomes an important factor increasing global mortality [1].

Moreover, one thing that cannot be neglected along with the increase of obese children is underweight. Underweight raises mortality at around a similar rate as obesity [4,5] and hinders children’s physical growth and development causing growth failure [6]. As such, energy imbalance during childhood may lead to limited growth, reduced fat-free mass (FFM), accumulated fat mass (FM), and greater risk of developing metabolic diseases [7,8,9]. Therefore, it is a highly significant issue for toddlers and children to maintain a normal weight.

As BMR (basal metabolic rate) plays a critical role in maintaining normal body weight as it takes about 60–80% of the entire energy consumption [10], it is important to carry out muscular exercises that boost FFM (fat-free mass) in the body to prevent obesity as FFM and BMR have a linear relationship, where greater FFM leads to higher BMR [11,12,13,14].

Muscular strength consists of health-related physical fitness, which includes cardiovascular endurance, muscular strength, endurance, flexibility, and body composition. They are related to diseases prevention and health enhancement [15]. In addition, as a research paper reported that a higher level of body growth, including height, increases muscular strength, muscular strength serves as a barometer for body growth [16]. Young children should try to maintain a high level of physical fitness [17,18], as physical fitness, including muscular strength, is used as an important health indicator for toddlers, children, and teenagers [17]. It is significant also because physical fitness built during one’s childhood greatly influences their physical fitness in adulthood and overall growth phase and continues onto their later life [19,20,21,22].

Furthermore, physical fitness greatly influences physical development and maintenance of a normal weight [23,24]. In children and teenagers, cardiovascular endurance, muscular strength, and muscular endurance have a significant relationship with development of cardiovascular metabolic diseases, adiposis, mental health, cognition, MSF (muscular strength fitness), and bone health [25,26,27]. Meanwhile, in the context of health improvement, attention should be paid to building physical fitness during one’s childhood as reports say that low cardiovascular endurance, muscular strength, and muscular endurance have a significant relationship with mortality [28,29,30].

Early childhood is an important period of physical development. During this period, crucial for building physical fitness, children can explore and experience physical movement through play, while learning and practicing exercise skills [31,32]. Inactive preschoolers have four times higher body fat compared to their active peers as they enter elementary school [33]. Moreover, when they turn six, fundamental motor skill (FMS)—locomotor, balance, and ball skills—enter maturation stage, where the physical ability obtained through various movement patterns becomes the foundation for physical fitness [34]. Therefore, it is highly important to study children aged six, as they grow from toddlers to children.

Evaluation is a process of finding out how effectively a program accomplishes educational goals [35], and physical fitness evaluation can be seen as the first step to achieving educational goals of physical activities [36]. Therefore, regular physical fitness evaluations with consistent, systematic exercise programs during one’s babyhood are critical to identify the current physical fitness status of children and seek supplementary ways if they fall short in certain aspects.

Hence, research on children’s physical fitness is steadily conducted, considering the importance of physical fitness built during babyhood. In such research, physical fitness is graded for evaluation [37,38]. As the phase critical for physical development during babyhood is irregular and the speed and level differ by individuals, the figures vary by their growth periods. Thus, it is difficult to measure the deviation of physical fitness by applying a unified gauge for physical fitness.

Therefore, this study analyzed the physique (physical development), physical fitness (muscular endurance, muscular strength, flexibility, agility, power, balance), and BMR of children aged six (73–84 months old). The measured physical fitness was then visually classified by physical fitness factors, and for children who showed different figures depending on their physical development and body composition, characteristics of clusters classified by the physique and BMR (precursor of FFM) were identified.

This clustering analysis on children’s physical fitness provided information on two things: first, the current physical status of the individual child and how much physical fitness the child lacked, and second, what effective exercise programs should be developed to supplement the physical fitness shortage of children in their growth stage and physical ability development. It could also provide necessary primary data for individual exercise programs that can counteract childhood obesity and underweight issues.

## 2. Materials and Methods

This study was conducted from September 2018 to February 2020.

### 2.1. Subjects

Subjects involved a total of 4410 children (2285 males, 2215 females) aged six (73–84 months old) residing in Seoul, Gyeonggi, Daegu, Jeonbuk, Inchoen, Chungnam, Daejeon, Ulsan, Jeju, Gyeongnam, and Gangwon.

### 2.2. Physical Growth Measurement

Height and weight were measured in 0.1 increments, the children were in light clothing while standing facing the front. The equipment used was DB-1H (CAS, Seoul, Korea).

### 2.3. Physical Fitness Measurement

For health-related physical fitness, subjects’ total body endurance, muscular strength, and flexibility were measured, modifying and supplementing the research tool used by Kim and Ann [39]. For skill-related physical fitness, power, balance, and agility were measured. The measuring equipment used by the Ministry of Education, Science, and Technology for children in the Physical Activity Promotion System was restructured for toddlers’ use [40].

#### 2.3.1. Muscular Endurance (Flexed Arm Hang)

The subjects hung on a bar above a mat with sensors attached for 60 s. The equipment measured the moment their feet touched the ground. The equipment used to measure muscular endurance was K.I.ME-2018 (Korea Insung, Seoul, Korea).

#### 2.3.2. Muscular Strength (Hand Grip Strength)

The subjects were told to stand at ease with arms straight down and squeeze the strengthener as hard as they could without bending their arm. The records were measured in 0.1 kg increments as the subjects gripped the strengthener one arm after another. The equipment used to measure muscular strength was T.K.K-1290 (Takei, Niigata, Japan).

#### 2.3.3. Flexibility (Sit and Reach)

The subjects were told to have their shoulders and hips attached to a bar, sit with legs straight and bend the lower back with both arms stretched forward. The records of how far the subjects pushed the bar were collected. For flexibility measurements, T.K.K-5826 (Takei, Niigata, Japan) was used.

#### 2.3.4. Power (Long Jump)

The subjects stood on the a with sensors attached, and how long they stayed in the air during a jump was converted into figures for records. They carried out two standing jumps, and the better record of the two was recorded for analysis. For power measurements, T.K.K-5414 (Takei, Niigata, Japan) was used.

#### 2.3.5. Agility (Shuttle Run)

The subjects were told to do a shuttle run, and a touch switch and timer were attached every 5 m to measure the records as they sprinted six times back and forth. For agility measurements, K.I.AG-2018 (Korea Insung, Seoul, Korea) was used.

#### 2.3.6. Balance (One Leg Stance Test)

For balance, the record of the time the subject stood on one leg with eyes open was measured. The subjects stood on a balance bar, stretched their arms wide sideways, bent their knees, slightly leaning forward. The recording was measured in 0.1 s increments. The figures were collected for each leg, and the measurement stopped once the balance broke and the leg touched the ground.

### 2.4. Obesity

This study set the obesity criteria based on the 2007 National Growth Charts for Children and Adolescents as follows: underweight, less than the 5th percentile of body weight by age; overweight, the body mass index by age more than the 85th percentile and less than the 95th percentile; and obesity, the 95th percentile or more. The 2007 National Growth Charts for Children and Adolescents is Korea’s standard growth chart made by the Korea Disease Control and Prevention Agency (KDCA) and the Korea Patriotic Society, adopting the World Health Organization (WHO) Growth Standards [41].

### 2.5. Basal Metabolic Rate (BMR)

The basal metabolism was calculated using the Oxford University equation for ages 3–10 [42]. The calculation formulas were: male BMR (MJ/day), 0.0937 × Wt + 2.15 and female BMR (MJ/day), 0.0842 × Wt + 2.12 (Wt: body weight).

### 2.6. Analysis Methods

#### 2.6.1. Clustering Analysis on Factors of Physical Fitness

Clustering is a method for finding a cluster structure in a data set that is characterized by the greatest similarity within the same cluster and the greatest dissimilarity between different clusters [43]. Clustering analysis is one of the unsupervised learning methodologies of machine learning that utilizes big data. It is an analysis method used to understand a cluster by subdividing observations into several groups. Groups of children that had similar figures in the six physical fitness factors were analyzed by clustering analysis. This research adopted the K-means algorithm suggested by MacQueen. The K-means algorithm first groups patterns into k clusters and calculates the average of patterns found in the clusters as the median value. Then, the distance between the median value of the cluster and each pattern is measured, after which the pattern is added to the closest cluster [44]. The six physical fitness factors used in the clustering analysis showed a wide variation in units and range, thus they were standardized as t-scores. Standardized values allow each variable to be equally contributing and placed at equal distances [45].

The study divided the subjects into three to seven clusters for analysis. Clusters with few samples were excluded (when they were subdivided into six to seven clusters). Then, after reviewing the characteristics of each cluster by two professionals in physical education for kids and physical fitness, the total number of clusters was reduced to four.

#### 2.6.2. Statistical Analysis Method

This study selected four clusters after conducting the K-means clustering analysis using IBM Statistics SPSS 25.0. The characteristics of the four selected clusters were then visualized and outlined. Then, the individual characteristics within the four clusters were analyzed through the chi-square test, after which one-way ANOVA was conducted to analyze physical development and basal metabolic aspects. The significance level for all statistical analyses was 0.05.

## 3. Results

### 3.1. Physical Health Cluster

As a result of the clustering analysis that investigated eight physical fitness measurement factors [muscular endurance, muscular strength (left, right), flexibility, agility, power, balance (left, right)], four physical fitness clusters could be found. Figure 1 is the octagonal chart of the four clusters and their physical fitness characteristics.

Cluster 1 showed a high performance in balance (left, right) and muscular endurance, while cluster 2 was outstanding in muscular strength (left, right) and muscular endurance. Cluster 3 appeared to show good agility performances while having lower results for the other five factors. Cluster 4 showed low figures in six factors. Therefore, it could be possible to summarize clusters 1–4 as a balance cluster, muscular strength cluster, low agility cluster, and low physical fitness cluster, respectively.

The composition ratio of the four physical fitness clusters and the skill statistics of the eight physical fitness factors (muscular endurance, muscular strength (left, right), flexibility, agility, power, balance (left, right)) are shown in Table 1. Cluster 4 was the highest proportion among the clusters with 35.2% of all case, followed by cluster 2 with 31.9% of all case, cluster 1 with 28.2% of all case, and cluster 3 with 4.8% of all case (Table 1).

### 3.2. Distributional Difference of Physical Growth Clusters per Individual (Physical Development) Characteristic

The results of the analysis conducted on the difference between physical fitness clusters per gender and physical development (Table 2) showed no significant difference in the distribution of physical fitness clusters by gender; however, a statistically significant difference was shown in the distribution of physical fitness clusters based on physical development.

A statistically significant difference appeared for all male and female children in terms of the distribution of physical development per cluster.

Male and female children with normal body weight accounted for approximately 30% in clusters 1 (balance), 2 (muscular strength), and 4 (low physical fitness). Meanwhile, the ratio of underweight children was high in clusters 1 (balance) and 4 (low physical fitness), while overweight and obese children accounted for a high percentage in clusters 2 (muscular strength) and 4 (low physical fitness).

As a result, the four clusters could be classified as 1—balance cluster, 2—muscular strength cluster, 3—low agility cluster, and 4—low physical fitness cluster. The low physical fitness cluster accounted for 35.2%, while it was 31.9% for the muscular strength cluster and 28.2% for the balance cluster. Many Korean children aged six turned out to belong in the low physical fitness cluster. As a result of analyzing the difference of physical fitness clusters by dividing the groups by gender, depending on their physical development, both genders showed differences in each group. For male children, those slim and slightly slim represented a high proportion in the balance cluster, while those slightly obese and obese were highly represented in the muscular strength cluster. For female children, those slightly slim and the standard group were highly represented in clusters 1 and 2, while many of those obese were found in cluster 2. Moreover, as a result of the variance analysis on physical development (height, weight) of each physical fitness cluster, in each group of male children and female children showed a statistically significant difference in height and body weight. They both in each group of male children and female children showed a high figure in height and weight in cluster 2 (muscular strength cluster). However, they showed various aspects in other clusters.

### 3.3. Differences in Physical Development by Physical Fitness Clusters

The results of the analysis conducted to identify the differences in physical development by physical fitness clusters are shown in Table 3. It turned out that the difference in both height and body weight was statistically significant. The muscular strength cluster (2) appeared to have the highest average figures of height (120.53 cm) and weight (24.66 kg) among all the clusters. The cluster with the lowest height average (116.62 cm) was the low agility cluster (3), while the balance cluster (1) had the lowest average body weight (21.99 kg).

For male children, the muscular strength cluster (2) had the highest average height (121.30 cm), while the figure was the lowest (117.75 cm) in the balance cluster. In terms of body weight, in the muscular strength cluster (2) children were the heaviest, with an average of 25.25 kg, while in the balance cluster (1), they were the lightest, with an average of 22.32 kg. For female children, in the muscular strength cluster (2) the average height was the highest (119.71 cm), while in the low agility cluster (3), it was the lowest (115.35 cm). In regard to body weight, the muscular strength cluster (2) showed the highest average (24.03 kg), while the balance cluster (1) the lowest (21.67 kg). In short, children in the muscular strength cluster (2) were found to have the best physical development status.

### 3.4. Differences in Basal Metabolic Rate (BMR) by Physical Fitness Clusters

The results of the analysis conducted to find out the physical fitness clusters’ differences in BMR are shown in Table 4. The differences in all four metabolic elements were statistically significant between the clusters. The muscular strength cluster showed the highest figure for BMR, while the lowest BMR was in the balance cluster. Both male and female children appeared to have the highest metabolic figures in the muscular strength cluster, while the lowest in the balance cluster.

## 4. Discussion

Physical development during the babyhood years shows a great variation as individuals grow at different speeds and levels over different phases. Therefore, it is insufficient to reflect the individual growth speed and physical fitness by grading their physical fitness on a fixed standard and measuring the differences. Thus, this study used a multivariate approach of physical fitness factors on the different physical development statuses of children, grouped the subjects into clusters, and analyzed their characteristics. This was also because a univariate analysis of physical fitness factors had room for biased interpretation of physical fitness.

As mentioned earlier, because children vary greatly in growth, an evaluation of their physical fitness through standardized grades, as in adults, would be limited. Therefore, physical fitness factors [muscular endurance, muscular strength (left, right), flexibility, agility, power, balance (left, right)] were visualized and clustered for measurement (Figure 1). This study classified the subjects into four physical fitness clusters. The balance cluster (1), where children had well-developed lower body muscles [46], accounted for 28.2% of all case. Meanwhile, cluster 2 consisted of children with balanced physical growth, particularly with developed upper body muscles, accounting for 31.9% of all case. The low agility cluster (3), where children had yet relatively undeveloped lower body muscles as they needed muscular strength to turn their body at a high speed [47], only accounted for 4.8% of all case. Cluster 4, characterized by low overall physical fitness, accounted for the largest proportion, 35.2% of all case. This implies that many Korean children have poor physical fitness, and that muscular strength and balance are important physical factors for their motor development. It can also be inferred that children begin developing agility from the age of 5.

It turned out that no difference was found between the physical fitness clusters of each gender, but there was a significant difference in terms of physical fitness depending on the children’s physique. The interpretation can be that male preschoolers did not differ by their physique from female preschoolers. This result can be supported by studies that reported that the BMI of elementary school students does not vary according to gender [48,49].

Such a result is considered stemmed from the different growth phases of the children; the differences in growth have led to differences in physical fitness. The reason why clusters are formed is thought to be due to the different development statuses of children depending on their physical fitness characteristics.

Muscular strength is a physical fitness element that steadily grows with age. The reason for the gradual increase of muscular strength is closely related to the growth of the body and the improvement of basic movement skills [49]. Muscular strength enables children to participate in physical activities where they can engage in various exercises, after which they can increase the quantity and difficulty of the physical activities [50]. Therefore, the research results imply that the muscular strength cluster consists of children that have more developed physical abilities than those in the other clusters, and that muscular strength plays a significant role in physical growth. It is mentioned as the most important physical fitness element for children’s growth, which once again emphasizes that it is a critical physical factor. Exercises that can help children boost muscular strength include jumping, which uses their body weight, and tug-of-war. In general, this so-called play-type physical activity aimed at children’s enjoyment is encouraged as the exercise to improve the muscular strength of children [34]. Muscular endurance, also one of the major physical fitness elements in babyhood, helps children make specific movements—such as throwing a ball, hanging onto a bar, shaking, climbing, and moving a heavy object—prevent injuries and keep a right posture.

In addition, as a high muscular endurance leads to greater performance in daily activities, plays, sports, and exercises [51], it can enlarge the range of physical activities and help children have a feeling of accomplishment. However, children who have not yet reached a sufficient stage of development are not stable in their ability to use the skills related to muscular endurance, so caution is required as they tend to use excessive power or contract their muscles more than necessary when performing muscle endurance-related movements [34].

Furthermore, balance can be defined as a process of maintaining the body’s center of gravity. One should steadily adjust one’s muscular activities and position of the joints [52]. In particular, static balance has a close relationship with the development of the lower body muscles (anterior tibial, soleus muscle, gastrocnemius muscle) [46] and is affected by visual and tactile–kinesthetic stimuli and stimulation of vestibular structures. According to Cratty and Martin [53], children aged six cannot balance with their eyes closed, but become able to do so as they turn seven. As this physical element improves steadily with age, it is a critical physical element that should be evaluated before the child enters school.

According to the results of this study, both male and female children that are slim or slightly slim had high figures in balance compared with their obese counterparts, which is probably due to the fact that accumulated adipose tissue and high body mass can hamper body balance [54]. Nevertheless, as low body weight during childhood and adolescence is related to reduced physical abilities, including muscular strength, muscular endurance, and cardiovascular abilities [55], it seems that other physical elements should be evaluated together with balance.

Agility refers to the ability to accurately move the entire body at a high speed [17]. Swift movements require decision-making cognitive ability and muscular strength. As growing children get older, their skeletal muscles and agility grow together [46]. This study found out that the lowest percentage of both male and female children was in the low agility cluster (3), which is probably due to the fact that decision-making cognitive ability and muscular strength for swift movements develop most lately during babyhood.

As we can predict that a lower BMR during babyhood leads to a higher BMI during the adolescent period, the cluster with relatively low BMR may start to gain weight at a later stage of childhood [56]. Therefore, it is highly important to evaluate the BMR of preschoolers. As people with high lean mass appear to have higher BMR [57], they may increase their BMR by building muscles through exercising.

As the study analyzed the differences in BMR between physical fitness clusters, both male and female children showed a statistically significant difference, and those in cluster 2 appeared to have the highest BMR. According to preceding research studying how the body and physical fitness affected BMR, involving about 500 children aged between seven and ten, grip strength and power affected BMR in male children, while only the grip strength was an affecting element in female children [58]. This study brought a similar result; the BMR of children in cluster 2, who had high muscular strength and muscular endurance, appeared to be the highest, and those cluster 1 had the lowest. As the BMR is influenced by body weight, cluster 1, which was mainly consisted of slim and slightly slim males and females, had the lowest BMR.

During babyhood i.e., before they enter elementary school, children grow rapidly in terms of not only physical abilities but also cognitive activities and abilities to protect themselves [59]. Therefore, the physical fitness evaluation of preschoolers is highly important as notable physical changes take place during that period. This study analyzed preschoolers by classifying their physique, physical development, and physical fitness, and found out that they varied greatly across the factors. This result implies that we need to take a more various evaluative approach toward assessing children’s physical fitness than dividing them into standardized grades. Cale and Harries emphasized the importance of physical activities—the process indicators—over consequent indicators, such as physical fitness or abilities [60]. As emotional satisfaction and happiness from being able to move the body freely as physical abilities, such as muscular endurance, muscular strength, flexibility, and balance, develop during babyhood have a positive effect on the formation of ego [48], it is necessary to establish and run exercise programs that involve various physical fitness factors. Hence, a multidirectional approach toward children’s physical fitness analysis is needed. It could also be used as the critical primary data for setting up physical fitness management programs customized to children’s physical fitness.

## 5. Conclusions

This study assessed the physical development, physical fitness (muscular endurance, muscular strength, flexibility, agility, power, balance), and BMR involving a total of 4410 children (2285 males, 2215 females) aged six (73–84 months old) residing in Korea. The physical fitness cluster could be subdivided into four clusters: balance (1), muscular strength (3), low agility (4), and low physical fitness (4). Cluster 1 showed a high ratio of slim and slightly slim children, while cluster 2 had a high proportion of children that were obese, tall, or heavy, and had the highest BMR. The study found a high deviation and diversity in the children’s physical fitness. Such a result implies that rather than grading the physical fitness elements through a univariate method, one should adopt a more various evaluative approach (univariate approach) to evaluate and look into physical activity programs. It can be considered that Korean children today have poorer physical fitness than in the past. The outcome of this study result may contribute to identifying individuals’ physical characteristics and carrying out customized physical activities. This will be able to help children grow up while building physical fitness, giving a healthy influence on the first stage of their life cycle.

## Figures and Tables

**Figure 1 children-08-00485-f001:**
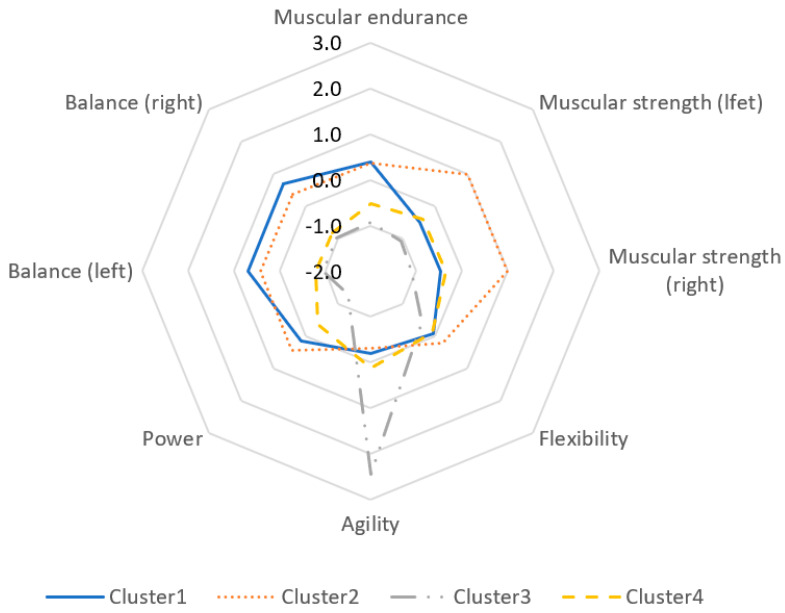
Physical fitness characteristics of each cluster.

**Table 1 children-08-00485-t001:** Performance statistics on eight physical fitness factors.

	Cluster No.	*n*	%	Muscular Endurance	Muscular Strength (Left)	Muscular Strength (Right)	Flexibility	Agility	Power	Balance (Left)	Balance (Right)
Total	1	1242	28.2%	41.73 ± 17.14	6.86 ± 1.65	7.30 ± 1.71	27.29 ± 7.04	13.3 ± 1.35	18.35 ± 3.92	48.37 ± 15.10	50.20 ± 14.31
	2	1406	31.9%	41.29 ± 17.58	10.51 ± 1.71	10.96 ± 1.71	29.36 ± 7.15	13.07 ± 1.43	19.6 ± 3.96	42.39 ± 17.85	43.54 ± 17.83
	3	210	4.80%	16.63 ± 16.35	5.52 ± 2.14	5.74 ± 2.22	24.66 ± 7.76	18.15 ± 3.20	12.28 ± 3.65	14.97 ± 14.28	15.87 ± 14.82
	4	1552	35.2%	24.50 ± 15.89	7.20 ± 2.00	7.64 ± 2.00	26.83 ± 7.72	13.88 ± 1.36	16.18 ± 3.59	18.11 ± 12.91	18.68 ± 13.26
Male	1	610	26.7%	42.53 ± 17.06	7.41 ± 1.59	7.86 ± 1.65	25.97 ± 7.26	13.02 ± 1.30	19.03 ± 4.01	45.83 ± 16.00	47.71 ± 15.17
	2	732	32.0%	42.80 ± 17.13	11.18 ± 1.62	11.55 ± 1.63	28.15 ± 7.00	12.75 ± 1.28	20.31 ± 4.09	39.76 ± 18.11	40.15 ± 18.36
	3	105	4.60%	14.84 ± 14.44	6.10 ± 2.16	6.10 ± 2.31	23.86 ± 7.77	18.24 ± 3.74	12.22 ± 4.03	13.78 ± 14.24	13.59 ± 12.21
	4	838	36.7%	25.53 ± 16.44	7.72 ± 1.98	8.11 ± 2.01	25.53 ± 7.67	13.58 ± 1.31	16.6 ± 3.7	15.82 ± 10.90	16.75 ± 12.08
Female	1	632	29.7%	0.97 ± 17.21	6.33 ± 1.52	6.77 ± 1.59	28.56 ± 6.58	13.58 ± 1.35	17.69 ± 3.71	50.82 ± 13.75	52.62 ± 13.00
	2	674	31.7%	39.64 ± 17.93	9.79 ± 1.50	10.32 ± 1.57	30.67 ± 7.07	13.43 ± 1.50	18.82 ± 3.68	45.25 ± 17.12	47.22 ± 16.48
	3	105	4.90%	18.43 ± 17.95	4.93 ± 1.96	5.38 ± 2.09	25.45 ± 7.71	18.07 ± 2.56	12.34 ± 3.25	16.16 ± 14.28	18.15 ± 16.79
	4	714	33.6%	23.29 ± 15.14	6.59 ± 1.85	7.09 ± 1.84	28.36 ± 7.50	14.23 ± 1.33	15.68 ± 3.39	20.8 ± 14.49	20.94 ± 14.20

1—balance cluster, 2—muscular strength cluster, 3—low agility cluster, 4—low physical fitness cluster.

**Table 2 children-08-00485-t002:** Cross-analysis of physical fitness clusters by individual characteristics.

			Cluster 1	Cluster 2	Cluster 3	Cluster 4	Total	Chi-Square	*p*
Gender		M	610 (26.7)	732 (32)	105 (4.6)	838 (36.7)	2285 (100)	6.894	0.075
		F	632 (29.7)	674 (31.7)	105 (4.9)	714 (33.6)	2125 (100)		
Physical Development	M	Underweight	20 (37)	2 (3.7)	7 (13)	25 (46.3)	54 (100)	83.511	0.000
		Normal	514 (30.3)	525 (30.9)	66 (3.9)	593 (34.9)	1698 (100)		
		Overweight	44 (15.5)	110 (38.9)	13 (4.6)	116 (41)	283 (100)		
		Obese	32 (12.8)	95 (38)	19 (7.6)	104 (41.6)	250 (100)		
	F	Underweight	21 (32.3)	8 (12.3)	7 (10.8)	29 (44.6)	65 (100)	88.522	0.000
		Normal	536 (34.2)	492 (31.4)	61 (3.9)	478 (30.5)	1567 (100)		
		Overweight	55 (16.4)	114 (33.9)	24 (7.1)	143 (42.6)	336 (100)		
		Obese	20 (12.7)	60 (38.2)	13 (8.3)	64 (40.8)	157 (100)		
Total	119 (2.7)	3265 (74)	619 (14)	407 (9.2)	4410 (100)				

*n* (%), cluster 1—balance cluster, cluster 2—muscular strength cluster, cluster 3—low agility cluster, cluster 4—low physical fitness cluster.

**Table 3 children-08-00485-t003:** Distributional analysis of physique (height, weight) by physical fitness clusters.

		Total				Male				Female			
		Mean	SD	F	*p*	Mean	SD	F	*p*	Mean	SD	F	*p*
Height	Cluster 1	117.30	4.31	124.247	<0.001	117.75	4.37	77.066	<0.001	116.86	4.22	50.582	<0.001
	Cluster 2	120.53	4.71			121.30	4.51			119.71	4.80		
	Cluster 3	116.62	6.26			117.90	6.20			115.35	6.10		
	Cluster 4	118.38	5.00			118.89	4.90			117.78	5.04		
	Total	118.68	4.98			119.31	4.92			118.00	4.96		
Weight	Cluster 1	21.99	3.22	115.906	<0.001	22.32	3.36	65.423	<0.001	21.67	3.04	49.691	<0.001
	Cluster 2	24.66	4.37			25.25	4.41			24.03	4.25		
	Cluster 3	23.37	5.49			24.18	6.07			22.55	4.73		
	Cluster 4	23.62	4.49			23.98	4.69			23.21	4.22		
	Total	23.48	4.31			23.95	4.49			22.98	4.05		

**Table 4 children-08-00485-t004:** Distributional analysis of metabolism by physical fitness clusters.

	Total				Male				Female			
	Mean	SD	F	*p*	Mean	SD	F	*p*	Mean	SD	F	*p*
Cluster 1	975.17	68.37	102.678	<0.001	1012.75	60.57	73.579	<0.001	938.90	54.50	52.214	<0.001
Cluster 2	1027.04	86.89			1068.11	76.36			982.44	74.84		
Cluster 3	995.58	106.48			1041.33	106.34			949.84	85.17		
Cluster 4	1006.40	87.52			1041.39	82.18			965.33	74.88		
Total	1003.67	85.80			1042.30	79.26			962.13	72.09		

cluster 1—balance cluster, cluster 2—muscular strength cluster, cluster 3—low agility cluster, cluster 4—low physical fitness cluster.

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
