# Peer review of "Analysis of Children’s Physical Characteristics Based on Clustering Analysis"

_children, 2021, doi:10.3390/children8060485_

Round 1

Reviewer 1 Report

Abstract

Line 12: Please consider to change “The study analyzed the relationship…… fitness

” to “The study identified the physical development, physical fitness……”

Introduction

Line 32: Please consider to reword “figure” to something like “number” throughout the paper.  

Line 35: I am not sure that obesity of toddler and children is related to early death. If it is true, author need to add references.

Line: 48: Needs to be spelled out first “BMR”

Line: 65: Needs to be spelled out first “MSF”

Line 77: please remove “pica, 2006”

Material and Methods

Please provide manufacturer, country, and etc. ? for all equipment.

2.1 Subject -> Please state when data was collected and how long did it take?

2.2 Physical growth measurement- >I would simply use height and weight

Please provide name of test author used for this study rather than muscular endurance, strength, flexibility, and power. Then explain what authors have done.

Ex) Flexibility: Sit and reach test was used to test flexibility……. The subject were told to have their…..

2.3.6 Balance

It is not clear kids performed balance test without eye open or with eye open. Further,

Line 150: It is sounds weird. “The records were measured in 0.1-sec increment”. Please consider to change.

2.5 For equation, please spell out the MJ and W.

2.6 Analysis Method

Please explain reason for using four clusters? I think it is important for this study.

Figure 1. Please add legends for clusters. Although authors define the cluster 1~4 later paragraph, it is difficult to understand by figure itself.

Tables: same as above. Please add legends for clusters. Ex) Cluster 1; balance cluster, Cluster 2; muscular strength cluster….

Line 295: Needs to be spelled out first “BMI”.

Line 292-295: This paragraph need to be rewritten. Further, I am not sure relationship between body growth, physical strength, and BMI.

Line 314-315: Please rewrite the sentence “In addition,………….Exercise [50].

Conclusion

Please consider to change the first sentence.

Line 384-387: The last sentence sounds weird. Please rewrite the sentence.

Author Response

Response to Reviewer 1 Comments

Thank you very much for reviewing our paper in detail.

Abstract

  1. Line 12: Please consider to change “The study analyzed the relationship…… fitness” to “The study identified the physical development, physical fitness……”

 → We revised the sentence again. “The study identified the physical development, physical fitness (muscular endurance, muscular strength, flexibility, agility, power, balance), and Basal Metabolic Rate (BMR) with a total of 4,410 children aged six (73-84 months) residing in Korea.”

Introduction

  1. Line 32: Please consider to reword “figure” to something like “number” throughout the paper. 

 →  Modified "figure" to "number".

  1. Line 35: I am not sure that obesity of toddler and children is related to early death. If it is true, author need to add references.

 → We additionally presented this previous study as a reference.

“In a study of approximately 7,000 obese children (ages 3-17) in Sweden and comparing them to approximately 34,000 children with the same conditions (age, gender, and area of residence), obese children had a greater risk of dying than children who did not. 36 %p increased.” 

[61] Lindberg, L.; Danielsson, P.; Persson, M., Marcus, C.; Hagman, E. Association of childhood obesity with risk of early all-cause and cause-specific mortality: A Swedish prospective cohort study. PLoS medicine, 2020, 17, e1003078.; DOI:10.1371/journal.pmed.1003078

  1. Line: 48: Needs to be spelled out first “BMR”

 → I modified it. BMR (basal metabolic rate)

  1. Line: 65: Needs to be spelled out first “MSF”

 → I modified it.  MSF (Muscular Strength Fitness) 

  1. Line 77: please remove “pica, 2006” 

 → I modified it. ( Line 87 : (pica, 2006”))

Material and Methods

  1. Please provide manufacturer, country, and etc. ? for all equipment.

 → We modified it as follows:

2.2. Physical growth measurement

Height and weight were measured in 0.1 units in a light-clad, frontal position. The equipment used to measure height and weight was DB-1H (CAS, Korea).

2.3.1. Muscular endurance

K.I.ME-2018, dead hang equipment made by Insung, a Korean company. → K.I.ME-2018 (Korea Insung,  Korea)

2.3.2. Muscular strength

 T.K.K-1290, a handgrip strengthener made by Takei, Japan. →  T.K.K-1290(Takei, Japan)

2.3.3. Flexibility

T.K.K-5826, equipment for seated pike stretching made by Takei, Japan, was used. → T.K.K-5826(Takei, Japan)

2.3.4. Power

T.K.K-5414, a standing jump measuring equipment made by Takei, Japan, was used.→ T.K.K-5414(Takei, Japan)

2.3.5. Agility

K.I.AG-2018 made by Insung, Korea, was used to measure agility. → K.I.AG-2018(Korea Insung, Korea)

  1. 1 Subject -> Please state when data was collected and how long did it take?

 → This study is a nationwide study, and data was collected for three semesters. We added content descriptions to the text as follows.

“ This study was conducted from September 2018 to February 2020.”

  1. 2 Physical growth measurement- >I would simply use height and weight

 → We have made the following simple amendments. 

“Height and weight were measured in 0.1 units in a light-clad, frontal position. The equipment used to measure height and weight was DB-1H (CAS, Korea).”

  1. Please provide name of test author used for this study rather than muscular endurance, strength, flexibility, and power. Then explain what authors have done. Ex) Flexibility: Sit and reach test was used to test flexibility……. The subject were told to have their…..

 → We revised the title as follows.

2.3.1. Muscular endurance (Flexed Arm Hang)

2.3.2. Muscular strength (Hand Grip Strengrh)

2.3.3. Flexibility (Sit and reach)

2.3.4. Power (Long jump)

2.3.5. Agility (shuttle run)

2.3.6. Balance (One Leg Stance Test)

  1. 3.6 Balance
    It is not clear kids performed balance test without eye open or with eye open.
    Further, Measurements were made with the eyes open. This content has been modified as follows.

“For balance, the records of how long the subjects stood on one foot were measured”  → “For balance, the record of the time the subject stood on one leg with  eyes open was measured.”

  1. Line 150: It is sounds weird. “The records were measured in 0.1-sec increments”. Please consider to change.

This content has been modified as follows.  

“The recording was measured in 0.1 second increments.”

  1. 5 For equation, please spell out the MJ and W.

→  We put BMR in front of the unit to help understand the MJ unit as shown below. I also added a description of  Wt.

“ Male BMR(MJ/day): 0.0937 × Wt + 2.15, Female BMR (MJ/day): 0.0842 × Wt + 2.12. (Wt: Body weight) ”

  1. 6 Analysis Method
    Please explain reason for using four clusters? I think it is important for this study. Figure 1. Please add legends for clusters. Although authors define the cluster 1~4 later paragraph, it is difficult to understand by figure itself.

→We inserted it at the bottom of 2.6.1 as shown below.

“The study divided the subjects into 3-7 clusters for analysis. Clusters with few samples were excluded (when they were subdivided into 6-7 clusters). Then, after reviewing the characteristics of each cluster with two professionals in physical education for kids and physical fitness, the total number of clusters was finalized to 4.”

  1. Tables: same as above. Please add legends for clusters. Ex) Cluster 1; balance cluster, Cluster 2; muscular strength cluster….

 → We added a description below the table.

 “cluster 1 (balance cluster), cluster 2 (muscular strength cluster), cluster 3 (agility cluster), cluster 4 (low physical strength cluster)” 

  1. Line 295: Needs to be spelled out first “BMI”.
    Line 292-295: This paragraph need to be rewritten. Further, I am not sure relationship between body growth, physical strength, and BMI.

We wanted to talk about the fact that there was no difference in physique by gender. I have modified the sentence as follows.

“where they stated that the same grade male and female elementary school students did not show a significant difference in physique(BMI).” 

It turned out that no difference was found between the physical fitness clusters of each gender, but there was a significant difference in terms of physical fitness depending on their physique. This can be interpreted that the physique of preschoolers does not differ by their gender. This result can be supported by the research that the physique (BMI) of elementary school students does not have a gender difference.

[48] Goodway JD, Ozmun JC, Gallahue DL. Understanding motor development: Infants, children, adolescents, adults; 8th edition; Jones & Bartlett Learning: Burlington, MA, 2019.

  1. ine 314-315: Please rewrite the sentence “In addition,………….Exercise [50].

We wrote the sentence again.

“In addition, as high muscular endurance leads to greater performance in daily activities, plays, sports, and exercises”

Conclusion

  1. lease consider to change the first sentence.
    Line 384-387: The last sentence sounds weird. Please rewrite the sentence.

→“The study identified the physical development, physical fitness~”   Changed as abstract

Once again, thank you for your careful review. I will try to do better research.

Thank you.

Reviewer 2 Report

Thanks for your work and opportunity to read what you have done. Really enjoyed it. 

Thank you for the opportunity to review this manuscript.

The content of the manuscript is within the scope of the Journal.

Below are my comments and suggested changes:

Abstractis good well done

line 31  Reference needed after 2017

line 41  Not sure that the link between obesity and underweight was made effectively. Suggest revision to have a smoother transition

line 74  In the first paragraph you have used 5 and then in this sentence you have used sixwould suggest consistency

line 92  You refer to physical strength but in the Abstract you referred to same measures as physical fitness would suggest consistency

line 106 Good sample size well done

line 110 Would like you to specify height e.g. is it standing height?

line 125 Please insert make and model of KIME 2018

line 131 Please insert make and model of TKK1290

line 136 Please insert make and model of TKK5826

line 141 Please insert make and model of TKK5414

line 145 Please insert make and model of KIAG2018

Section 2.3.1 to 2.3.5 read a little repetitive Would suggest revision to aid in readability and interest. I also would like to see you establish the reliability of the apparatus that you have used in your methods. Are they reliable at collecting what you say they collect?

line 173 You are referring to physical strength again in Abstract you said physical fitness. Suggest changes so that it is consistent throughout you entire document.

line 167 Maybe a comment of the advantages of cluster analysis here. Why did you choose to cluster? Maybe even in the Introduction would be nice to establish the argument.

Results are looking good well done

I would really like to see a discussion around the practical applications of your findings. How would a practitioner use your results to improve child health outcomes?

I would like to see a discussion around the limitations of your research. I think this is an important part of your work.

References look goodthank you

Author Response

Response to Reviewer 2 Comments

Review2

Comments and Suggestions for Authors

Thanks for your work and opportunity to read what you have done. Really enjoyed it.

 Thank you for the opportunity to review this manuscript.

The content of the manuscript is within the scope of the Journal.

 Below are my comments and suggested changes:

Thank you for your careful and sincere review. We tried to make our research valuable. Thank you.

Abstractis good –well done

  1. Line 31  Reference needed after 2017 

 → [1] It is the same as the bibliography. Add.

  1. line 41  Not sure that the link between obesity and underweight was made effectively. Suggest revision to have a smoother transition

 → We additionally presented this previous study as a reference.

“In a study of approximately 7,000 obese children (ages 3-17) in Sweden and comparing them to approximately 34,000 children with the same conditions (age, gender, and area of residence), obese children had a greater risk of dying than children who did not. 36 %p increased.” 

[61] Lindberg, L.; Danielsson, P.; Persson, M., Marcus, C.; Hagman, E. Association of childhood obesity with risk of early all-cause and cause-specific mortality: A Swedish prospective cohort study. PLoS medicine, 2020, 17, e1003078.; DOI:10.1371/journal.pmed.1003078

  1. line 74  In the first paragraph you have used 5 and then in this sentence you have used six–would suggest consistency

→Preschool is under 7 years of age and studies are conducted by each age group. We made it under the age of six, but the papers we referred to varied in age groups.

  1. line 92  You refer to physical strength but in the Abstract you referred to same measures as physical fitness –would suggest consistency

→ We decided "physical fitness" and modified it.

  1. line 106 Good sample size –well done

→ Thank you.

  1. line 110 Would like you to specify height e.g. is it standing height?

→ We modified it as follows.

“Height and weight were measured in light clothing while standing facing the front.  It was measured in 0.1 units, and the equipment used was DB-1H (CAS, Korea).

  1. line 125 Please insert make and model of KIME 2018
    line 131 Please insert make and model of TKK1290
    line 136 Please insert make and model of TKK5826
    line 141 Please insert make and model of TKK5414
    line 145 Please insert make and model of KIAG2018

→ We modified it as follows.

2.3.1. Muscular endurance

K.I.ME-2018, dead hang equipment made by Insung, a Korean company. → K.I.ME-2018 (Korea Insung,  Korea)

2.3.2. Muscular strength

 T.K.K-1290, a handgrip strengthener made by Takei, Japan. →  T.K.K-1290(Takei, Japan)

2.3.3. Flexibility

T.K.K-5826, equipment for seated pike stretching made by Takei, Japan, was used. → T.K.K-5826(Takei, Japan)

2.3.4. Power

T.K.K-5414, a standing jump measuring equipment made by Takei, Japan, was used.→ T.K.K-5414(Takei, Japan)

2.3.5. Agility

K.I.AG-2018 made by Insung, Korea, was used to measure agility. → K.I.AG-2018(Korea Insung, Korea)

  1. Section 2.3.1 to 2.3.5 read a little repetitive Would suggest revision to aid in readability and interest. I also would like to see you establish the reliability of the apparatus that you have used in your methods. Are they reliable at collecting what you say they collect?

→ Our research tools are validated in the paper below. It is also described in the text as the preceding study [39] of the research tool. The description of the research tool has been modified a little bit.

An. N.Y., Lee, S.N., Park, T.S. Kim. E.J. A Fundamental Study on the Development of the Guidelines for Healthy Living Habits and Physical Activities of children. Journal of Converging Sport and Exercise Sciences, 2020, 18, 27-44.

Kim. E.J., An. N.Y., Comparison of Infant Health Factors Related to Stress (α-Amylase) - A Basic Study for Development of Child Health Behavioral Model, The Korean Journal of Growth and Development, 2021, 29, 163-171.

  1. line 173 You are referring to physical strength again –in Abstract you said physical fitness. Suggest changes so that it is consistent throughout you entire document.

→ We decided "physical fitness" and modified it.

  1. line 167 Maybe a comment of the advantages of cluster analysis here. Why did you choose to cluster? Maybe even in the Introduction would be nice to establish the argument.

→ In the introduction, we explained the reasons for cluster analysis as follows.

“This clustering analysis on children’s physical fitness provides information on two things: first, the current physical status of the individual child and how much physical fitness the child lacks, and second, developing effective exercise programs that can supplement the physical fitness shortage of children in their growth stage and physical ability development. It could also provide necessary primary data for individual exercise programs that can improve childhood obesity and underweight issues.”

In the following, the cluster analysis of this study was further explained.

  1. Results are looking good –well done

→Thank you for the compliment.

  1. I would really like to see a discussion around the practical applications of your findings. How would a practitioner use your results to improve child health outcomes? I would like to see a discussion around the limitations of your research. I think this is an important part of your work.

→Thank you for your good advice. We supplemented the discussion as follows.

“Cluster 1 was the balance cluster, where children had well-developed lower body muscles [45] accounting for 28.2%. Meanwhile, cluster 2 consisted of children with balanced physical growth, particularly with developed upper body muscles, accounting for 31.9%. Cluster 3 was the low agility cluster where children had yet relatively undeveloped lower body muscles as they need muscular strength to turn their body at a high speed [46]. This cluster only accounted for 4.8%. Cluster 4 had low overall physical fitness, accounting for the largest proportion of 35.2%. This implies that many Korean children have poor physical fitness and that muscular strength and balance are important physical factors for their motor development. Also, it can be inferred that children begin developing agility from the age of 5.”

  1. References look good–thank you

 →Thank you

Once again, thank you for your careful review. Your good advice will help you study for life. Thank you so much.

It's a tough day with COVID 19.

Reviewer, take care of your health and be happy.
